# The Different Temozolomide Effects on Tumorigenesis Mechanisms of Pediatric Glioblastoma PBT24 and SF8628 Cell Tumor in CAM Model and on Cells In Vitro

**DOI:** 10.3390/ijms23042001

**Published:** 2022-02-11

**Authors:** Eligija Damanskienė, Ingrida Balnytė, Angelija Valančiūtė, Marta Maria Alonso, Aidanas Preikšaitis, Donatas Stakišaitis

**Affiliations:** 1Department of Histology and Embryology, Medical Academy, Lithuanian University of Health Sciences, 44307 Kaunas, Lithuania; ingrida.balnyte@lsmuni.lt (I.B.); angelija.valanciute@lsmuni.lt (A.V.); 2Department of Pediatrics, Clínica Universidad de Navarra, University of Navarra, 31008 Pamplona, Spain; mmalonso@unav.es; 3Centre of Neurosurgery, Clinic of Neurology and Neurosurgery, Faculty of Medicine, Vilnius University, 03101 Vilnius, Lithuania; danas911@gmail.com; 4Laboratory of Molecular Oncology, National Cancer Institute, 08660 Vilnius, Lithuania

**Keywords:** pediatric glioblastoma, temozolomide, NKCC1, KCC2, EZH2, PCNA, CAM

## Abstract

It is necessary to elucidate the individual effects of temozolomide (TMZ) on carcinogenesis and tumor resistance to chemotherapy mechanisms. The study aimed to investigate the TMZ 50 and 100 μM dose effect difference between PBT24 and SF8628 cell line high-grade pediatric glioblastoma (phGBM) xenografts in a chicken chorioallantoic membrane (CAM) model, on PCNA and EZH2 immunohistochemical expression in the tumor and on the expression of NKCC1, KCC2, E- and N-cadherin genes in TMZ-treated and control cell groups in vitro. TMZ at a 100 μg dose reduced the incidence of PBT24 xenograft invasion into the CAM, CAM thickening and the number of blood vessels in the CAM (*p* < 0.05), but did not affect the SF8628 tumor in the CAM model. The TMZ impact on PBT24 and SF8628 tumor PCNA expression was similarly significantly effective but did not alter EZH2 expression in the studied tumors. The TMZ at 50 μM caused significantly increased RNA expression of the NKCC1 gene in both studied cell types compared with controls (*p* < 0.05). The expression of the KCC2 gene was increased in PBT24 TMZ-treated cells (*p* < 0.05), and no TMZ effect was found in SF8628-treated cells. The study supports the suggestion that individual sensitivity to TMZ should be assessed when starting treatment.

## 1. Introduction

Pediatric high-grade glioblastoma multiforme (phGBM) is a highly malignant brain tumor and the most common cause of death [1]. Five-year survival in phGBM patients is less than 20 percent [2]. The standard glioblastoma treatment includes surgical resection, radiotherapy and temozolomide (TMZ) chemotherapy [3]. Following therapy with TMZ in adults, the treatment has similarly employed TMZ for phGBM patients [2,4]. Notwithstanding, patients having an initial response to TMZ fail therapy: approximately 55% of glioblastoma patients develop resistance to TMZ chemotherapy [5,6]. phGBM is different from adult gliomas. The unique developmental origins and distinct biological factors of this heterogeneous group of tumors have highlighted the importance of avoiding treatment strategies based solely on adult glioblastoma, as this approach has not improved the outcome of phGBM [7]. Individual TMZ effectiveness depends on the resistance to TMZ, which would cause glioblastoma recurrence and a worse outcome [8]. It is essential to determine individual sensitivity to TMZ treatment and the personal effect of TMZ on cancerogenesis, which is critical for effective treatment.

The main antitumor effect of anticancer medicines is the inhibition of tumor cell proliferation and the promotion of apoptosis. In glioblastoma, markers of tumor progression include the expression of proliferating cell nuclear antigen (PCNA), enhancer of zeste homolog 2 (EZH2) and ionic transporters that regulate intracellular chloride levels [9,10,11]. PCNA is an important target for many aggressive tumors. The proliferation of tumor cells is correlated with a high degree of tumor malignancy, which can be evaluated by measuring the PCNA protein expression [12]. Thus, the potential for targeting PCNA in chemotherapy against aggressive tumors is actively pursued [13]. Researchers have reported Polycomb repressive complex 2 (PRC2) activity in phGBM [14]. PRC2 is usually accompanied by cancer progression [15]. PRC2 helps to support gene silencing and X chromosome inactivation through the enzymatic methylation of K27 on histone H3 by EZH2. EZH2 is a catalytic PRC2 subunit, a histone methyltransferase targeting lysine 27 of histone H3 [16]. This methylated H3-K27 consequence is associated with the silencing of different genes in phGBM [14]. Functional interconnections among EZH2-mediated histone methylation and DNA methylation indicate the gene silencing involved in the loss of tumor suppression [9].

High expression of the Na-K-2Cl co-transporter (NKCC1) in glioblastoma is involved in cell proliferation [11]. The sign of apoptosis is a reduction in cell volume arising from a loss of intracellular K^+^([K^+^]i) and chloride ([Cl^−^]i) [10,17]. The increased [Cl^−^]i level in glioblastoma cells is related to upregulated NKCC1 and downregulated K-Cl co-transporter (KCC2) [18,19]. Increased NKCC1 protein expression in human glioblastoma directly correlates with the tumor grade and cell migration; NKCC1 inhibition reduces glioblastoma cell migration and tumor invasion [20,21]. The repressing effect on glioma cell migration is expected to result from reduced [Cl^−^]i [22]. Knockdown of NKCC1 in glioblastoma cells causes the formation of significantly more extensive focal adhesions and cell traction forces than in control cells [20]. Blockade of the NKCC1 protein function serves as a therapeutic strategy to overcome TMZ-mediated glioblastoma resistance [23].

In adult glioblastoma epileptic patients, a reduction in KCC2 neuropil staining [19] and a drop in [K^+^]i and [Cl^−^]i concentrations in the TMZ-treated glioma cells were described [24,25]; a loss of K^+^ and Cl^−^ that occurred through apoptosis was confirmed [26]. Thus, it is necessary to investigate whether TMZ initiates KCC2 activity in glioblastoma cells, resulting in the parallel loss of K^+^ and Cl^−^ ions.

High expression of NKCC1 is related to epithelial–mesenchymal transition (EMT) in gliomas, offering a new therapeutic strategy for inhibiting tumor progression [27]. The EMT process is linked with resistance to treatment, and EMT in glioblastoma cells may be complemented by enhanced N-cadherin (CDH2) expression, correlating with adverse prognosis [28]. The association of phGBM progression with cadherin-E (CDH1) and CDH2 is mainly unexplored, whereas the literature on the association of CDH1 and CDH2 expression with adult glioma progression is controversial [29,30,31]. TMZ generates a DNA O6-methylguanine lesion that triggers DNA restoration, drains the enzyme O6-methylguanine methyltransferase and begins glioblastoma cell apoptosis, produced by activating external apoptotic or mitochondrial-dependent pathways [32]. TMZ causes cell arrest in the G2/M cell cycle [33] and reduces cell proliferation during exposure to medicine [33,34]. Despite this, TMZ arouses NKCC1 expression and activity in glioblastoma cells [24].

The chicken chorioallantoic membrane (CAM) model clarifies investigational medicines for cancer treatment and is an alternative model to study tumor growth, invasion and angiogenesis [35,36]. The CAM model, being immunodeficient, allows transplantations from different tumor tissues and species without immune responses [37]. The CAM model has advantages over the rodent immunodeficiency models in that it is cheap, allows monitoring of the CAM epithelium basement membrane barrier disruption by the tumor, induces neo-angiogenesis, permits the detection of drug effects and has a short-term duration, which can be helpful in the prediction of anticancer therapy’s efficacy [38]. Our study demonstrated that the spheroid model did not reflect the treatment effect on tumor progression observed in the CAM model [39].

The present study aimed to investigate the differences between a 3-year-old girl’s high-grade SF8628 cell line xenograft and a 13-year-old boy‘s high-grade PBT24 cell line [40] xenograft using the CAM model; to examine the tumor response to treatment with TMZ’s effect on PCNA and EZH2 histological expression in cells of the tumor in the CAM; and to investigate the impact of TMZ on the NKCC1, KCC2, E- and N-cadherin gene expression in the studied phGBM cells.

## 2. Results

### 2.1. The Biomicroscopy of PBT24 and SF8628 Xenograft on CAM

Figure 1 shows stereomicroscopic images of the PBT24 and SF8628 control and TMZ-treated tumor on the CAM at 5 days of chick embryo development (EDD9—day 2, EDD12—day 5 post-transplantation), extracted EDD12 CAM with tumor (ex ovo) and their histological view (H–E). On EDD9, the PBT24 and SF8628 control tumors were larger than on EDD12, indicating that the tumor invaded into the CAM mesenchyme three days later, and only the superficial part of the tumor was visible above the membrane. The PBT24 tumor on EDD9 was dense with precise edges. On EDD12, the PBT24 control and 50 µM TMZ-treated tumors were without clear borders, appearing smaller due to a smaller tumor on the membrane, as part of the tumor invaded the CAM (Figure 1; EDD9, EDD12, ex ovo, and H–E), and the vascular network (“spoked-wheel”) formed around the tumor (Figure 1). Figure 2 shows the vascular network formed around the control and 50 µM TMZ-treated tumor, obviously visible after the injection of fluorescent dextran into the CAM vessel. The PBT24-100 µM TMZ tumor EDD12 was dense, with a clear border, growing on the surface of the CAM, with muted neo-angiogenesis (Figure 1 and Figure 2).

The stereomicroscopically visible SF8628 control EDD12 tumor was smaller than the TMZ-treated tumors, and the histological H–E images show the profoundly invasive nature of the tumor growth (Figure 1). Compared to EDD9, the SF8628 TMZ-treated EDD12 tumor size was less pronounced as the tumor grew and was predominantly exposed on the CAM surface or was shallowly invasive into the CAM mesenchyme. There was an apparent “spoked-wheel” vascular network around the SF8628 control EDD12 and 50 µM TMZ-treated tumor, which was less pronounced around the 100 µM TMZ-treated tumor (Figure 1 and Figure 2).

### 2.2. The PBT24 and SF8628 Growth, Invasion into CAM Frequency, the CAM Thickness and the Number of Blood Vessels in CAM under the Tumors of the Study Groups

Compared to the control, 100 µM TMZ reduced the frequency of PBT24 tumor invasion into the CAM (*p* = 0.007), while 50 µM TMZ had no effect on invasion (*p* > 0.05). A significant difference was found when the PBT24-50 µM TMZ and PBT24-100 µM TMZ groups were compared (*p* = 0.019). Compared to the SF8628 control at EDD12, treatment of the SF8628 tumor with TMZ did not reduce the tumor invasion frequency into the CAM in the SF8628-treated groups (*p* > 0.05; Table 1; Figure 3).

Compared with the PBT24 control on EDD12, the CAM thickness beneath the tumor was significantly lower in the PBT24-100 µM TMZ (*p* = 0.0003). The PBT24-100 µM TMZ CAM thickness was lower than in the PBT24-50 µM TMZ group (*p* = 0.0009). Treatment of the SF8628 tumor with both doses of TMZ did not affect the CAM thickness under the EDD12 tumor (*p* > 0.05; Table 1).

When comparing the neo-angiogenesis expression on EDD12 of the control in the CAM with that in the PBT24-treated TMZ groups, both doses of TMZ significantly inhibited angiogenesis in the CAM under the PBT24 tumor (*p* < 0.002), but there was no difference in this parameter between the TMZ-treated groups (*p* > 0.05). The treatment with TMZ had no suppressive effect on neo-angiogenesis in the SF8628 tumor groups (*p* > 0.05; Table 1).

### 2.3. The PCNA and EZH2 Expression in PBT24 and SF8628 Tumors

Table 2 and Figure 4 and Figure 5 show the PCNA and EZH2 positively stained cell expression in the tissue of the studied tumors at EDD12.

A significantly higher number of PCNA-positive cells was found in the SF8628 control than in PBT24 control tumors (*p* < 0.05; Figure 5a). The effect of TMZ treatment on the expression of the studied markers was similar in PBT24 and SF8628 tumors. Compared to the control, both doses of TMZ significantly reduced the number of PCNA-positive cells in PBT24 and SF8628 tumors (*p* < 0.05), but the 100 µM TMZ dose was significantly more effective compared to the 50 µM TMZ dose (Table 2; Figure 5a).

A significantly higher level of EZH2-positive cells was detected in SF8628 control tumors than in PBT24 control tumors (*p* < 0.05). No significant reduction in EZH2-positive cells was observed in TMZ-treated tumor tissue compared to the matched control or when comparing the expression of EZH2-positive cells between the treated groups (*p* > 0.05; Table 2; Figure 5b).

### 2.4. The Expression of SLC12A2 (NKCC1 Co-Transporter) and SLC12A5 (KCC2 Co-Transporter) Gene in PBT24 and SF8628 Cell Study Groups

The expression of *SLC12A2* in PBT24 and SF8628 cell groups is shown in Table 3 and Figure 6. We found no difference in *SLC12A2* expression between the PBT24 and SF8628 control groups. Treatment with 50 µM TMZ significantly increased *SLC12A2* expression in PBT24 (*p* = 0.0022) and SF8628 cells (*p* = 0.0022). The effect of 50 µM TMZ on *SLC12A2* expression was significantly lower in SF8628-50 µM TMZ than in PBT24-50 µM TMZ cells (Figure 6a,c; Table 3). Compared to the control, the 50 µM TMZ dose increased *SLC12A2* expression in PBT24 cells by two-fold (2^−ΔΔCT^ = 2.04) and in SF8628 cells by 1.5-fold (2^−ΔΔCT^ = 1.5).

The expression of the *SLC12A5* and *GAPDH* genes and the differences in the expression found when comparing the groups studied are shown in Table 3. The expression of *SLC12A5* in SF8628 control cells was significantly lower than in the PBT24 control (*p* = 0.0022). The TMZ dose of 50 µM increased *SLC12A5* expression 2.6-fold (2^−ΔΔCT^ = 2.6) in PBT24 cells and 1.6-fold in SF8628 cells compared to the respective control.

The 50 µM dose of TMZ significantly increased *SLC12A5* expression in PBT24 cells (*p* = 0.0022), and the treatment had no significant effect on the gene expression in SF8628 cells. There was significantly lower expression of *SLC12A5* in the SF8628-50 µM TMZ group compared to the PBT24-50 µM TMZ group (*p* = 0.0022; Figure 6b,d; Table 3).

Comparison of the mean value of the ΔCT *SLC12A5*/ΔCT *SLC12A2* ratio of PBT24 and SF8628 cells showed the significantly higher ratio value of SF8628 (4.70 (4.27–4.81) compared to the PBT24 control (3.87 (2.95–4.17); *p* < 0.002). The treatment with TMZ increased the ΔCT *SLC12A5*/ΔCT *SLC12A2* value in PBT24 cells to 4.50 (4.3–5.1) (*p* < 0.003), and in SF8628 cells to 5.26 (5.05–5.58) (*p* < 0.003). When comparing the ratio value between PBT24- and SF8628-treated cell groups, it was significantly higher in the SF8628-50 µM TMZ than in the PBT24-treated group (*p* < 0.009; Figure 7).

The correlation (*r*) between *SLC12A2* and *SLC12A5* ΔCT values was 0.71 in control PBT24 cells, 0.14 in control SF8628 cells, 0.66 in treated PBT24 and 0.49 in treated SF8628 cells (*p* > 0.05 in all groups).

### 2.5. The Expression of CDH1 (E-Cadherin) and CDH2 (N-Cadherin) Gene in PBT24 and SF8628 Cell Study Groups

The expression of *CDH1*, *CDH2* and *GAPDH* genes and the differences between the study groups are shown in Table 4. Significantly higher *CDH1* expression in the PBT24 control cells than in the SF8628 control group was found (*p* = 0.0022). This shows the more invasive phenotype of SF8628 cells. *CDH1* expression in the PBT24-50 µM TMZ group was also higher than in the SF8628-50 µM TMZ group (*p* = 0.0022). No differences in *CDH2* expression were found when comparing PBT24 cells with SF8628 control groups (*p* > 0.05), but *CDH2* expression in the PBT24-50 µM TMZ group was found to be significantly higher than that in SF8628 cells treated with 50 µM TMZ (*p* = 0.0022; Figure 8a,b; Table 4).

Treatment with 50 µM TMZ did not significantly affect *CDH1* and *CDH2* expression compared to controls in either treated group (*p* > 0.05; Table 4). The 50 µM dose of TMZ increased *CDH1* expression in PBT24 cells by two-fold (2^−ΔΔCT^ = 2.0) and *CDH2* expression by 1.5-fold, while SF8628 cells had a 0.8-fold decrease in *CDH1* expression and a 1.16-fold increase in *CDH2* expression compared to their controls (Figure 8c,d).

## 3. Discussion

In a previous study, TMZ chemotherapy significantly improved overall survival in the elderly group but had a more limited effect in the younger group [41]. The ineffectiveness of glioblastoma treatment in the face of a high level of glioblastoma polymorphism has shown that targeting all patients with a single strategy is unrealistic to achieve treatment progress. Thus, a personalized pharmacological therapy for glioblastoma should be tailored to the individual patient’s tumor pathophysiological, molecular, genetic and gender-related characteristics [42]. Our experimental study using a CAM model demonstrates the differences in the efficacy of TMZ therapy and the associated molecular mechanisms in the treatment of pediatric PBT24 and SF6828 tumors. The study found that TMZ at a dose of 100 µM significantly reduced the incidence of PBT24 tumor invasion into the CAM and the thickness of the CAM, and significantly inhibited neo-angiogenesis in the CAM beneath the PBT24 tumor, but had no effect on the SF8628 tumor growth and the corresponding parameters studied.

Cancer cell proliferation may involve a non-oncogenic structural protein, such as PCNA, which acts as a “hub” for large cellular complexes that is essential for tumor growth and cancer cell survival. Drugs reducing PCNA expression in tumor cells are expected to have a broader anticancer therapeutic spectrum than medicines targeting specific signal proteins [43]. The control SF8628 tumor had significantly more PCNA-positive cells than the PBT24 control. The effect of TMZ treatment on PCNA expression in PBT24 and SF8628 tumors was similar and dose-dependent. PCNA is involved in DNA metabolic processes, including DNA replication and repair, chromatin organization and transcription. PCNA is necessary for cell metabolic processes such as glycolysis [13].

The study found a significantly higher level of EZH2-positive cells in the SF8628 control tumor than in the PBT24 control. EZH2 is commonly overexpressed in glioblastoma and is firmly associated with tumor malignancy [44,45,46]. It is essential to assess the impact of the medicine integration on EZH2 blockade when explaining phGBM therapy strategies [15]. The inhibition of EZH2 reverses TMZ chemosensitivity in glioblastoma [47]. However, no significant reduction in EZH2-positive cells was observed in TMZ-treated PBT24 and SF8628 tumor tissue compared to their controls in our study.

Cancer progression is related to Cl^−^ and Na^+^ in the tumor microenvironment [11,48,49]. The persistence of high neuronal levels of NKCC1 in pediatric glioblastoma supports the hypothesis of abnormal and immature neuronal cells in the phGBM. Strong NKCC1 immune reactivity in the aberrant neuronal component of glioblastoma and no upregulation of neuronal NKCC1 was observed in the perilesional area of tumor specimens [19]. Our study found no difference in *SLC12A2* expression between control PBT24 and SF8628 groups. Treatment with 50 µM TMZ significantly increased *SLC12A2* expression in PBT24 and SF8628 cells compared to controls in PBT24 cells by 2-fold and in SF8628 cells by 1.5-fold. Increased expression of NKCC1 protein and its elevated phosphorylation, with a concurrent increase in the phosphorylation of serine–threonine kinases WNK, in TMZ-treated glioblastoma was reported [24,50,51]. The researchers suggested that NKCC1 activity in TMZ-treated cells was stimulated via Cl^−^/volume-sensitive regulatory kinases and the WNK-mediated signaling pathway, which is vital in protecting glioma from a loss of cell volume and Cl^−^ during TMZ treatment. The regulatory WNK kinases, a family of serine–threonine kinases, are activated by losing [Cl^−^]i and cell shrinkage [51,52]. The rapid upregulation of these proteins is likely due to de novo protein synthesis using mRNA reserves, allowing the glioblastoma cells to adapt instantaneously to the altered osmotic situation [24].

Therapeutic resistance has been proposed to emerge from the overexpression of the NKCC1 transporter, which intensifies DNA repair mechanisms against TMZ-induced apoptosis [24]. Inhibition of NKCC1 activity by bumetadine accelerates TMZ-treated glioblastoma cell apoptosis, and this suggests that NKCC1 activity remains functional and further regulates cell volume in TMZ-treated glioma, playing a role in [Cl^−^]i supplementation [24].

High-grade glioblastoma cells accumulate intracellular chloride ([Cl^−^]i) to ~10-fold higher levels compared with the average in grade II glioma and the normal cortex [20]. It was proposed that some factors could dilute K^+^ and Cl^−^ concentrations in TMZ-treated cells. Researchers reported that aquaporin 4 protein was downregulated in glioblastoma cells after chemotherapy and radiotherapy, with reduced peritumoral brain edema [53]. Silencing WNK kinase activity can promote Na-K-2Cl inhibition and K-Cl co-transporter activation via net transporter dephosphorylation, revealing WNKs’ ability to modulate [Cl^−^]i [50]. Apoptosis requires persistent cell shrinkage and loss of cell volume via the reduction of [K^+^]i and [Cl^−^]i, which occurs before any other detectable apoptosis features [25,26,54]. The study data show that SF8628 control cells have significantly lower KCC2 gene (*SLC12A5*) expression than PBT24 cells. TMZ treatment significantly increased *SLC12A5* expression in PBT24 cells, while treatment of SF8628 cells had no significant effect on gene expression. Moreover, *SLC12A5* expression was substantially lower in the SF8628-treated TMZ than in the PBT24-treated group.

Adult glioblastoma patients with epilepsy syndrome showed a decrease in KCC2 staining in tumor tissue [19] and a reduction in [K^+^]i and [Cl^−^]i levels in TMZ-treated glioma cells [24,25]. Loss of [K^+^]i and [Cl^−^]i in the glioma cell in parallel with expressed apoptosis was confirmed [26]. Therefore, it is important to carry out further research to determine whether the distinct effect of TMZ in stimulating KCC2 activity in glioblastoma cells is due to a patient-specific impact of TMZ in promoting KCC2 activity in cells and the relationship of this with the efficacy of tumor treatment.

The study data of the ΔCT *SLC12A5*/ΔCT *SLC12A2* ratio in PBT24 and SF8628 cells show that the ratio value of the SF8628 control cells is significantly higher than that of PBT24. Treatment with TMZ significantly increased the value of ΔCT *SLC12A5*/ΔCT *SLC12A2* in both TMZ-treated groups, and when comparing the value between the PBT24- and SF8628-treated groups, it was substantially higher in the SF8628 group. Additionally, these data suggest that the efficacy of TMZ treatment may be related to changes in [Cl^−^]i, with a Cl^−^ concentration increase in SF8628 cells associated with increased NKCC1 gene expression and no modifications of KCC2 gene expression. In contrast, PBT24-treated cells showed an apparent rise in KCC2 gene expression, with a lower value of the co-transporter gene ratio, which may have led to a decrease in K^+^ and Cl^−^ concentrations in TMZ-treated PBT24 cells. The reduction of the intracellular K^+^ and Cl^−^ ion levels is related to the activation of caspases and triggers caspase cascade-related apoptosis mechanisms [17]. The decline of intracellular K^+^, Na^+^ and Cl^−^ results in an 80–85% loss of cell volume, DNA degradation and apoptotic body development in Jurkat cells [25].

Reactive astrocytes express NKCC1 in glioblastoma [19]. NKCC1 upregulation may lead to astrocyte swelling [55,56] and produce a GABAA receptor-mediated excitatory response, facilitating seizures [57,58,59]. The paradoxical excitatory action of GABAA depends on the relatively high [Cl^−^]i content in the cell [58]. On the other hand, KCC2 is a neuron-specific Cl^−^ extruder that uses a K^+^ gradient to maintain a low [Cl^−^]i level to ensure the proper functioning of postsynaptic GABAA receptors. Studies over the last two decades have shown that low KCC2 activity results in excitatory GABAergic transmission associated with seizures. KCC2 expression and function are features of epileptic disorders in the developing and adult brain. The effect of drugs that activate KCC2 function in glioblastoma is important as a potential new therapeutic target for treating glioblastoma [60]. Future studies of the colocalization of Cl^−^ co-transporters with the GABAA receptor may shed light on the importance of the functional interaction of Cl^−^ transporters in glioblastoma cells. In our study, statistically significantly higher expression of *CDH1* was detected in PBT24 control cells compared to SF8628. Researchers have shown that a decrease in *CDH1* expression is associated with astrocytoma progression [29,30], while other studies showed that high E-cadherin expression is associated with a poorer prognosis of the disease [31]. The contribution of E-cadherin expression to adult glioblastoma and phGBM progression remains unclear. No differences were found when comparing the expression of *CDH2* in PBT24 cells with that in the SF8628 control. Treatment with TMZ had no statistically significant effect on *CDH1* and *CDH2* expression.

Treatment with TMZ was found to be effective in inhibiting PBT24 tumor growth on the CAM and its invasion into the CAM, and inhibiting neo-angiogenesis, but was ineffective on the SF8628 xenograft. This difference is possibly related to TMZ’s differential effect on the carcinogenesis mechanisms regulating [Cl^−^]i levels, where PBT24 cells showed initially higher KCC2 expression, and its activation by TMZ therapy. It cannot be excluded that the found differences among cell lines are also related to sex-specific disparities. Sex-specific analyses can improve accuracy in identifying the molecular subtype of glioblastoma, and patients can achieve a better outcome by personalizing treatment according to sex differences in molecular mechanisms [61].

## 4. Materials and Methods

### 4.1. Cell Lines and Cell Culture

A 13-year-old boy’s high-grade glioblastoma PBT24 cell line cells were donated by Prof. M. M. Alonso (University of Navarra, Spain) [40] for the study. A 3-year-old girl’s diffuse intrinsic pontine glioblastoma (DIPG) SF8628 cell line cells—harboring the histone H3.3 Lys 27-to-methionine (Sigma Aldrich, St. Louis, MO, USA)—were also studied [62,63]. The PBT24 cells were cultivated in Roswell Park Memorial Institute 1640 (RPMI) medium (Sigma Aldrich, St. Louis, MO, USA). The media were supplemented with 10% fetal bovine serum (FBS; Sigma Aldrich, St. Louis, MO, USA) containing 100 IU/mL of penicillin and 100 µg/mL of streptomycin (P/S; Sigma Aldrich, St. Louis, MO, USA). The SF8628 cells were cultivated in Dulbecco’s Modified Eagle Medium (DMEM)–High-Glucose media (Sigma Aldrich, St. Louis, MO, USA). The media were supplemented with 10% fetal bovine serum (FBS; Sigma Aldrich, St. Louis, MO, USA) containing 100 IU/mL of penicillin and 100 µg/mL of streptomycin (P/S; Sigma Aldrich, St. Louis, MO, USA) and 2 mM L-Glutamine (Sigma Aldrich, St. Louis, MO, USA). Cells were incubated at 37 °C in a humidified 5% CO_2_ atmosphere.

### 4.2. The CAM Model

According to the legislation in force in the EU and Lithuania, no approval for studies using the CAM model is needed from the Ethics Committee. Cobb 500 fertilized chicken eggs were obtained from a local hatchery (Rumšiškės, Lithuania) and kept in an incubator (Maino incubators, Oltrona di San Mamette, Italy) at 37 °C temperature and 60% relative air humidity. The eggs were rolled automatically once per hour until the third embryo development day (EDD3).

The CAM was detached from the eggshell at EDD3; the eggshell was cleaned with 70% ethanol, a small round hole was drilled in the location of the air chamber, and approximately 2 mL of the egg white was aspirated. A window of approximately 1 cm^2^ in the eggshell was drilled and sealed with sterile transparent plastic tape. The eggs were kept in the incubator without rotation until GB cell tumor grafting on CAM at the seventh embryo development day (EDD7).

### 4.3. The PBT24 and SF8628 Tumor Study Groups

The growth and invasion into the CAM of the formatted PBT24 cell, as well as of SF8628 cell line xenografts, were investigated in the 6 groups. The study groups were as follows: PBT24-control (*n* = 13), PBT24-50 µM TMZ (*n* = 13), PBT24-100 µM TMZ (*n* = 10). The studied SF8628 tumor groups were the following: SF8628-control (*n* = 13), SF8628-50 µM TMZ (*n* = 14), SF8628-100 µM TMZ (*n* = 13).

Biomicroscopy in vivo and histological analyses of invasion, the thickness of the CAM and the number of vessels in the CAM under the tumor were performed.

The immunohistochemical (IHC) expression of PCNA in the tumor was studied in the following groups: PBT24-control (*n* = 9), PBT24-50 µM TMZ (*n* = 6), PBT24-100 µM TMZ (*n* = 6), SF8628-control (*n* = 8), SF8628-50 µM TMZ (*n* = 6), SF8628-100 µM TMZ (*n* = 7).

The expression of the EZH2 was investigated in the following: PBT24-control (*n* = 6), PBT24-50 µM TMZ (*n* = 7), PBT24-100 µM TMZ (*n* = 7), SF8628-control (*n* = 8), SF8628-50 µM TMZ (*n* = 6), SF8628-100 µM TMZ (*n* = 6). Efficacy studies of TMZ on GB in vivo and in vitro study at selected 100 and 50 μM doses were based on our and other investigators’ data [64].

### 4.4. Biomicroscopy Data to Assess Tumor Growth and Drug Efficacy

The biomicroscopy of xenografts on CAM at embryo development from 9 to 12 days (EDD9–12) in vivo is suitable for evaluating the tumor growth characteristics and its malignancy, and detecting the disparities among different cell line tumors and the sensitivity to treatment. One sign of tumor malignancy and growth progression is the relatively rapid formation of vasculature around the tumor—a “spoked-wheel” consisting of tumor-attracted small blood vessels and formed by neo-angiogenesis new blood vessels. The tumor size, border clarity and changes in the “spoked-wheel” expression may serve as features of the drug effect on tumorigenesis.

### 4.5. Tumor Grafting on CAM In Vivo

An absorbable gelatin surgical sponge (Surgispon, Aegis Lifesciences, India) was cut manually with a blade into pieces of 9 mm^3^ (3 × 3 × 1 mm). The 1 × 10^6^ cells were resuspended in 20 µL of rat tail collagen, type I (Gibco, New York, NY, USA) (in the control group), and temozolomide (TMZ; Sigma Aldrich, St. Louis, MO, USA) dissolved in dimethyl sulfoxide (DMSO; Sigma Aldrich, St. Louis, MO, USA). A 20 µL liquid mixture of tumor cells was pipetted onto a piece of sponge. The 50 µM TMZ- and 100 µM TMZ-treated tumor groups and their controls were formed. At EDD7, the tumor was grafted onto the CAM among significant blood vessels. Its structural changes were observed with a stereomicroscope (SZX2-RFA16, Olympus, Tokyo, Japan) in vivo during the EDD9–12 period. The tumor images were acquired using a digital camera (DP92, Olympus, Tokyo, Japan) and CellSens Dimension 1.9 digital imaging software.

### 4.6. Histological Study of the Tumor

At EDD12, the specimens were harvested, fixed in a buffered 10% formalin solution for 24 h and embedded in paraffin wax. The tumor sample was cut with a microtome (Leica, Nussloch, Germany) into 3-µm-thick sections. The sections were stained with H–E and IHC methods. Visualization and photographing of H–E- and IHC-stained tumor slides were performed using a light microscope (BX40F4, Olympus, Tokyo, Japan) and a digital camera (XC30, Olympus, Tokyo, Japan) equipped with CellSens Dimension 1.9 software.

H–E-stained tumors were divided into two types: invasive and non-invasive. The tumor invasion into the CAM was categorized as the destruction of the chorionic epithelium (ChE) or/and tumor cell invasion into the CAM mesenchyme. The tumor not invaded into mesenchyme was located on the CAM surface, and the chorionic epithelium’s integrity was not disrupted. The tumor invasion was examined in H–E slides at 20× and 40× magnifications.

### 4.7. Assessment of the CAM Thickness and the Number of Blood Vessels in CAM

The CAM thickness (width) was evaluated by photographing H–E-stained CAM at 4× magnification directly under the tumor. The thickness of CAM was measured (µm) in ten areas. The median CAM thickness was calculated in the area under the tumor.

The number of blood vessels was assessed by photographing the H–E-stained CAM at 4× magnification directly under the tumor. Blood vessels larger than 10 µm were counted.

### 4.8. Immunohistochemical Study

The expression of the PCNA and EZH2 markers was determined in tumor cells by immunohistochemistry. Primary antibodies to PCNA (PC10, Thermo Fisher Scientific, Branchburg, NJ, USA) and KMT6/EZH2 (phospho S21, ab84989, Abcam, Cambridge, UK) were used to detect PCNA and EZH2 positively stained tumor cells. Thin CAM sections of 3 µm were mounted onto adhesion slides (Thermo Fisher Scientific, Branchburg, NJ, USA), deparaffinized and rehydrated by standard techniques. Heat-induced antigen retrieval was performed using a Tris/EDTA buffer at pH 9 (K8002, Dako, Glostrup, Denmark) and a pressure cooker at 95 °C for 20 min (Thermo Fisher Scientific, Branchburg, NJ, USA). The Shandon CoverPlate System (Thermo Fisher Scientific, Branchburg, NJ, USA) was used for staining. Endogenous peroxidase was blocked with the Peroxidase Blocking Reagent (SM801, Dako, Glostrup, Denmark). The slides were treated with primary antibodies (1:100) for 30 min at room temperature. The primary antibody and antigen complex was determined using the horseradish peroxidase-labeled polymer dextran conjugated with a secondary mouse antibody and a linker (SM802 and SM804, respectively; Dako, Glostrup, Denmark) for 30 min at room temperature. Positive reactions were visualized using the 3,3′-diaminobenzidine-containing chromogen (DAB, DM827, Dako, Glostrup, Denmark), which gives a brown color to the site of the target antigen recognized by the primary antibody. After each step, a Tris-buffered saline solution containing Tween 20 (DM831, Dako, Glostrup, Denmark) was used as a wash buffer. Slides were counterstained with the Mayer hematoxylin solution (Sigma Aldrich, Taufkirchen, Germany), dehydrated, cleared and mounted.

For assessment of the tumor PCNA and EZH2 protein expression, two random vision fields (plot area 23,863.74 µm^2^) of the immunohistochemically stained tumor were photographed at 40× magnification. All cells and the PCNA and EZH2 positively stained cells were calculated in selected vision fields, and the percentages of PCNA- and EZH2-positive cells were counted in each tumor.

### 4.9. Extraction of RNA from PBT24 and SF8628 Cell Line Cells

PBT24 and SF8628 cell line cells were treated with 50 µMTMZ for 24 h. The concentration of 50 µM was chosen because it corresponds to the mean plasma concentration of the drug in TMZ-treated patients [65]. Control groups were cultured in a cell culture medium depending on the cell line. According to the manufacturer’s instructions, the total RNA was extracted using the TRIzol Plus RNA Purification Kit (Life Technologies, New York, NY, USA). The RNA quality and concentration were assessed using a NanoDrop2000 spectrophotometer (Thermo Scientific, Branchburg, NJ, USA). The total RNA integrity was analyzed using the Agilent 2100 Bioanalyzer system (Agilent Technologies, Santa Clara, CA, USA) with an Agilent RNA 6000 Nano Kit (Agilent Technologies, Santa Clara, CA, USA). The samples of RNA were stored at −80 °C until further analysis.

### 4.10. Determination of the SLC12A5, SLC12A2, CDH1 and CDH2 Gene Expression in PBT24 and SF8628 Cell Line Cells

RNA expression assays were performed for *SLC12A5* (Hs00221168_m1), SLC12A2(Hs00169032_m1), *CDH1* (Hs01023894_m1), *CDH2* (Hs00983056_m1) and *GAPDH* (Hs02786624_g1) genes. According to the manufacturer’s instructions, reverse transcription was performed with the High-Capacity cDNA Reverse Transcription Kit with RNase Inhibitor (Applied Biosystems, Waltham, MA, USA) in a 20 µL reaction volume containing 50 ng RNA using the Biometra TAdvanced thermal cycler (Analytik Jena AG, Jena, Germany). The synthesized copy DNA (cDNA) was stored at 4 °C until use or at −80 °C for a longer time. Real-time polymerase chain reaction (PCR) was performed using an Applied Biosystems 7900 Fast Real-Time PCR System (Applied Biosystems, Waltham, MA, USA). The reactions were run in triplicate with 4 µL of cDNA template in a 20 µL reaction volume (10 µL of TaqMan Universal Master Mix II, no UNG (Applied Biosystems, Waltham, MA, USA), 1 µL of TaqMan Gene Expression Assay 20× (Applied Biosystems, Waltham, MA, USA), 5 µL of nuclease-free water (Invitrogen, Carlsbad, CA, USA)), with the program running at 95 °C for 10 min, followed by 45 cycles of 95 °C for 15 s, 50 °C for 2 min and 60 °C for 1 min.

The control and 24 h TMZ-treated groups (*n* = 6 per group) were tested for *SLC12A5*, *SLC12A2*, *CDH1* and *CDH2* expression.

### 4.11. Statistical Analysis

The statistical analysis was performed using the Statistical Package for Social Sciences, version 23.0 for Windows (IBM SPSS Statistics V23.0). The frequency of tumor invasion into the CAM was expressed as a percentage (%), and the chi-square test was used to compare tumor invasion into CAM frequency between the study groups. The Shapiro–Wilk test was used to verify the normality assumption. Data of PCNA and EZH2 positively stained cells, the number of blood vessels and the CAM thickness are expressed as median and range (minimum and maximum values). The difference between the two independent groups was evaluated using the nonparametric Mann–Whitney U test.

To investigate the KCC2, NKCC1, E-cadherin and N-cadherin genes RNA expression in the TMZ-treated and control groups, the threshold cycle (CT) value was normalized with the control *GAPDH*, and the ΔCT value was obtained. The Livak method (∆∆CT) was used for calculating the relative fold change in expression levels [66]. The Spearman’s rank correlation coefficient (*r*) was used to assess relationships between the *SLC12A5* and *SLC12A2* (ΔCT values were used). Differences at the value of *p* < 0.05 were considered significant. The figures were created using GraphPad Prism 7 and IBM SPSS Statistics 23.0 software.

## 5. Conclusions

PCNA and EZH2 marker assays of PBT24 and SF8628 glioblastoma tumors transplanted on CAM showed that the PBT24 tumor is less aggressive than the SF8628 tumor. TMZ treatment effectively decreased PBT24 xenograft growth but did not affect the SF8628 tumor. TMZ treatment reduced PCNA expression in PBT24 and SF8628 tumors and had no effect on EZH2 expression. TMZ activated Na-K-2Cl co-transporter gene expression in both tumors but increased K-Cl co-transporter gene expression only in PBT24 cells. The efficacy of the treatment may be related to changes in intracellular Cl^−^ levels induced by TMZ exposure. These data highlight the importance of studies on the activity of the K-Cl co-transporter in the context of personalized anticancer therapy efficacy.

## Figures and Tables

**Figure 1 ijms-23-02001-f001:**
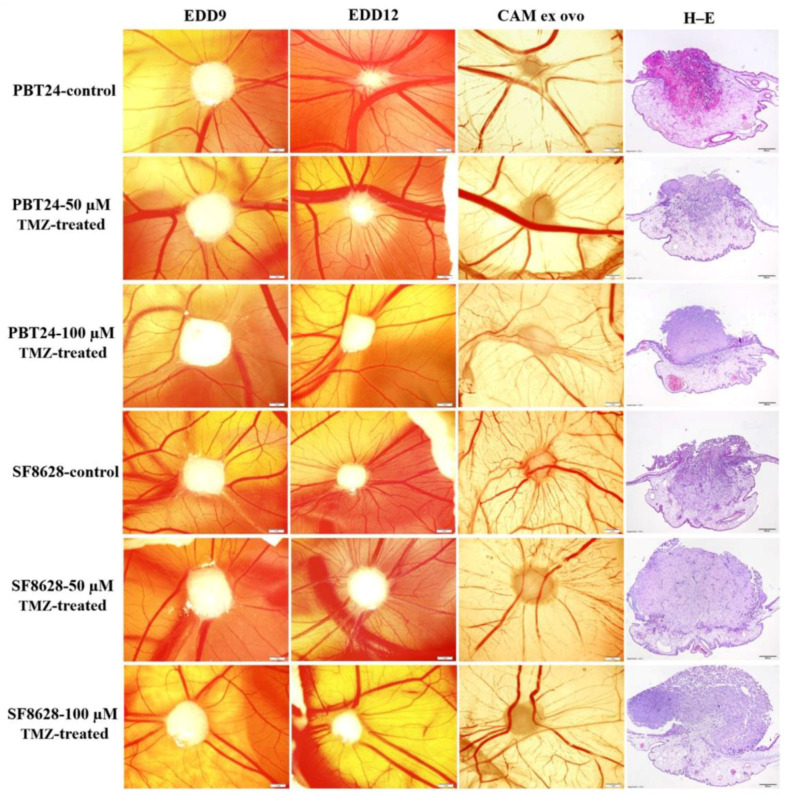
Stereomicroscopy of PBT24 and SF8628 tumors in vivo, a chorioallantoic membrane with tumor ex ovo and the histologic images of the study groups. EDD9, EDD12 and CAM ex ovo scale bar—1 mm; hematoxylin and eosin (H–E) stained preparations‘ scale bar—200 µm.

**Figure 2 ijms-23-02001-f002:**
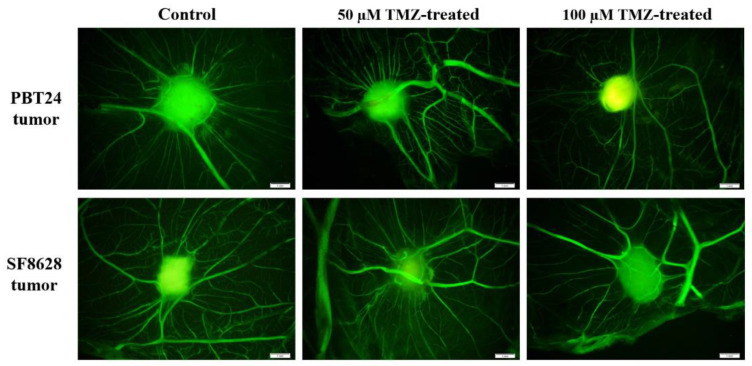
Fluorescent stereomicroscopy assay with fluorescent dextran of PBT24 and SF8628 tumors. Dextran highlighted the tumor and vascular network around it. Scale bar—1 mm.

**Figure 3 ijms-23-02001-f003:**
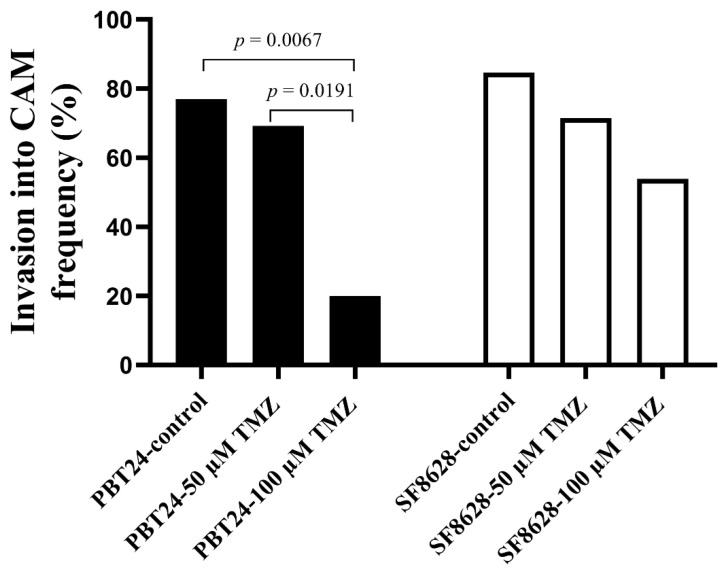
PBT24 and SF8628 tumor invasion into CAM frequency in control and TMZ-treated groups.

**Figure 4 ijms-23-02001-f004:**
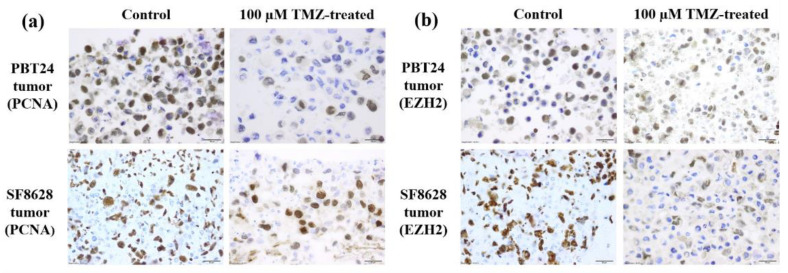
PCNA and EZH2 positively stained tumors of the PBT24 and SF8628 control and 100 µM TMZ-treated study groups. Dark brown nuclei indicate a PCNA-positive (**a**) and EZH2-positive cell (**b**). Scale bar—20 µm.

**Figure 5 ijms-23-02001-f005:**
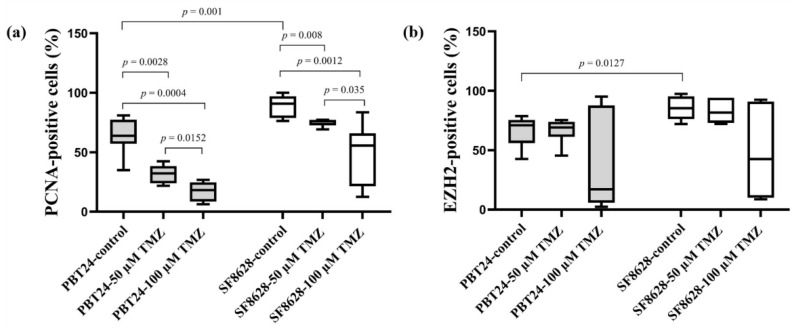
The percentage of PCNA-positive (**a**) and EZH2-positive (**b**) cells in PBT24 and SF8628 tumors.

**Figure 6 ijms-23-02001-f006:**
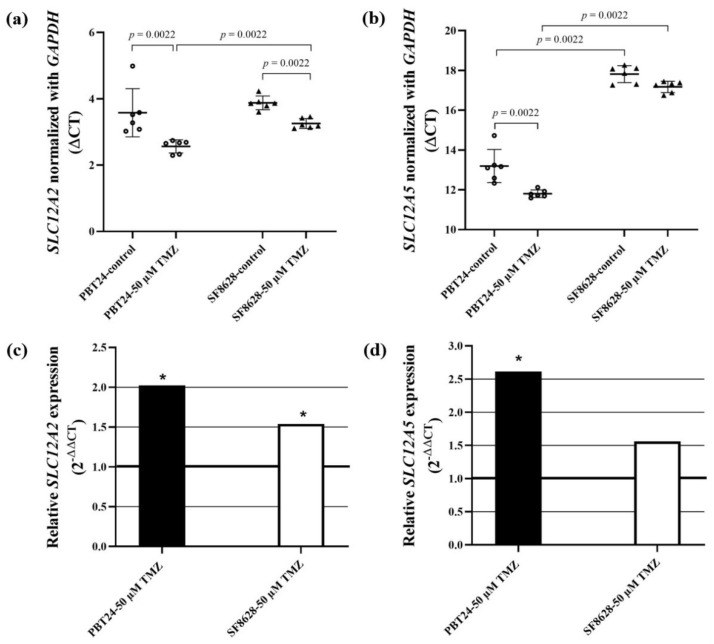
*SLC12A2* (**a**) and *SLC12A5* (**b**) expression in PBT24 and SF8628 control groups and 50 µM TMZ-treated groups. Data are after normalization with the *GAPDH* gene. Delta threshold cycle (ΔCT) values were used for the graph (the horizontal bars represent the mean; the short horizontal lines show standard deviation (SD) values). *SLC12A2* (**c**) and *SLC12A5* (**d**) relative expression in PBT24 and SF8628 50 µM TMZ-treated groups. The relative gene expression in TMZ-treated groups compared with respective controls. The 1.0 line shows the starting point of gene expression; * *p* < 0.05.

**Figure 7 ijms-23-02001-f007:**
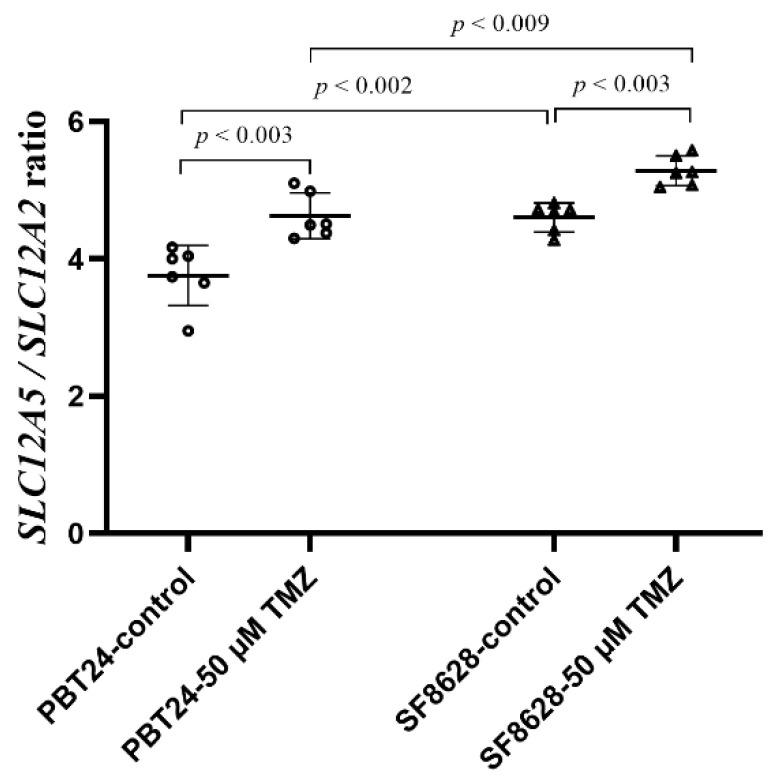
Comparison of the mean of ΔCT *SLC12A5/*ΔCT *SLC12A2* ratio value with SD among the PBT24 and SF8628 cell study groups.

**Figure 8 ijms-23-02001-f008:**
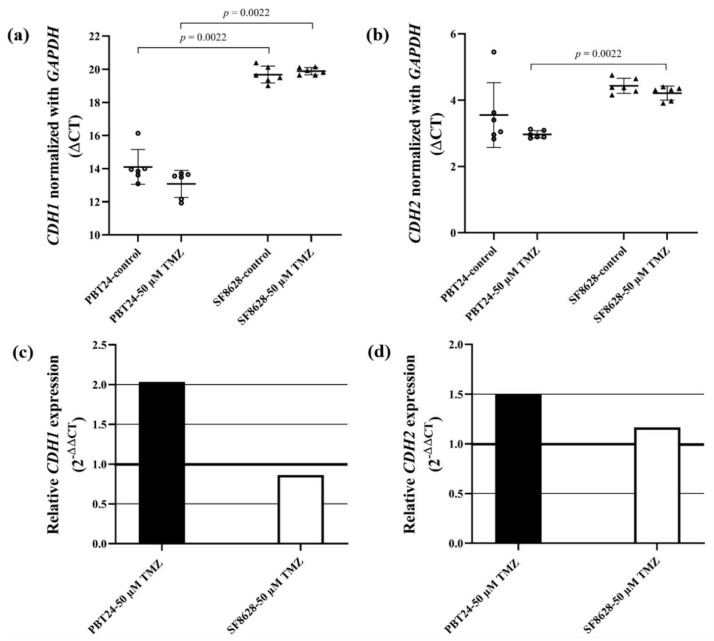
*CDH1* (**a**) and *CDH2* (**b**) expression in PBT24 and SF8628 control groups and 50 µM TMZ-treated groups. Data are after normalization with *GAPDH*. Delta threshold cycle (ΔCT) values were used for the graph (the horizontal bars represent the mean; the short horizontal lines show SD values). *CDH1* (**c**) and *CDH2* (**d**) gene expression in PBT24 and SF8628 50 µM TMZ-treated groups. The relative gene expression in TMZ-treated groups as compared with their control groups. The 1.0 line shows the starting point of gene expression.

**Table 1 ijms-23-02001-t001:** PBT24 and SF8628 tumor invasion into CAM frequency, the thickness of the CAM and the number of blood vessels in the CAM under the tumors of the study groups.

Study Group	*n*	Invasion Frequency (%)	CAM Thickness (µm)	Number of Blood Vessels
			Median (Range)	
PBT24-control	13	76.92	300.88 (65.23–700.87)	15 (6–28)
PBT24-50 µM TMZ	13	69.23	321.39 (67.02–516.85)	9 (3–14) ^e^
PBT24-100 µM TMZ	10	20.0 ^a,b^	55.48 (38.4–275.2) ^c,d^	5.5 (3–13) ^f^
SF8628-control	13	84.61	282.5 (47.85–539.7)	15 (5–21)
SF8628-50 µM TMZ	14	71.43	419.4 (84.49–683.7)	15 (5–29)
SF8628-100 µM TMZ	13	53.85	252.1 (55.51–529.1)	14 (7–19)

^a^*p* = 0.007, compared with PBT24-control; ^b^
*p* = 0.0191, compared with PBT24-50 µM TMZ; ^c^
*p* = 0.0009, compared with PBT24-50 µM TMZ; ^d^
*p* = 0.0003, compared with PBT24-control; ^e^
*p* = 0.0012, compared with PBT24-control; ^f^
*p* = 0.0001, compared with PBT24-control.

**Table 2 ijms-23-02001-t002:** The percentage of PCNA and EZH2 positively stained cells in PBT24 and SF8628 tumors of the study groups.

Study Group	PCNA-Positive Cells (%)		EZH2-Positive Cells (%)	
	*n*	Median (Range)	*n*	Median (Range)
PBT24-control	9	63.78 (34.87–80.95)	6	71.00 (42.63–78.70)
PBT24-50 µM TMZ	6	32.12 (21.78–42.42) ^a^	7	69.15 (45.38–75.37)
PBT24-100 µM TMZ	6	18.15 (6.25–26.80) ^b,c^	7	17.11 (2.38–95.06)
SF8628-control	8	90.81 (76.27–100) ^d^	8	85.36 (72.04–97.45) ^h^
SF8628-50 µM TMZ	6	76.17 (69.19–77.28) ^e^	6	81.71 (72.13–94.08)
SF8628-100 µM TMZ	7	55.65 (12.45–83.57) ^f,g^	6	42.55 (8.68–92.45)

^a^*p* = 0.0028, compared with PBT24-control; ^b^
*p* = 0.0004, compared with PBT24-control; ^c^
*p* = 0.0152, compared with PBT24-50 µM TMZ; ^d^
*p* = 0.001, compared with PBT24-control; ^e^
*p* = 0.0080, compared with SF8628-control; ^f^
*p* = 0.0012, compared with SF8628-control; ^g^
*p* = 0.0350, compared with SF8628-50 µM TMZ; ^h^
*p* = 0.0127, compared with PBT24-control.

**Table 3 ijms-23-02001-t003:** RNA expression of *SLC12A2* and *SLC12A5* gene in PBT24 and SF8628 cell study groups.

Study Group	*n*	CT Mean		ΔCT Mean ± SD	ΔΔCT
		*SLC12A2*	*GAPDH*		
PBT24-control	6	22.951	19.372	3.579 ± 0.73	
PBT24-50 µM TMZ	6	21.766	19.214	2.552 ± 0.2 ^a^	−1.027
SF8628-control	6	22.894	19.017	3.876 ± 0.21	
SF8628-50 µM TMZ	6	22.215	18.966	3.249 ± 0.15 ^b,c^	−0.628
		** *SLC12A5* **	** *GAPDH* **	**ΔCT mean ± SD**	**ΔΔCT**
PBT24-control	6	32.564	19.372	13.191 ± 0.83	
PBT24-50 µM TMZ	6	31.047	19.214	11.833 ± 0.19 ^d^	−1.359
SF8628-control	6	36.831	19.017	17.814 ± 0.43 ^e^	
SF8628-50 µM TMZ	6	36.127	18.966	17.161 ± 0.29 ^f^	−0.652

^a^*p* = 0.0022, compared with PBT24-control (*SLC12A2*); ^b^
*p* = 0.0022, compared with PBT24-50 µM TMZ (*SLC12A2*); ^c^
*p* = 0.0022, compared with SF8628-control (*SLC12A2*); ^d^
*p* = 0.0022, compared with PBT24-control (*SLC12A5*); ^e^
*p* = 0.0022, compared with PBT24-control (*SLC12A5*); ^f^
*p* = 0.0022, compared with PBT24-50 µM TMZ (*SLC12A5*).

**Table 4 ijms-23-02001-t004:** RNA expression of E- and N-cadherin in PBT24 and SF8628 cell study groups.

Study Group	*n*	CT Mean		ΔCT Mean ± SD	ΔΔCT
		*CDH1*	*GAPDH*		
PBT24-control	6	33.476	19.372	14.104 ± 1.05	
PBT24-50 µM TMZ	6	32.294	19.214	13.079 ± 0.81	−1.024
SF8628-control	6	38.689	19.017	19.672 ± 0.51 ^a^	
SF8628-50 µM TMZ	6	38.851	18.966	19.885 ± 0.22 ^b^	0.213
		** *CDH2* **	** *GAPDH* **	**ΔCT mean ± SD**	**ΔΔCT**
PBT24-control	6	22.924	19.372	3.552 ± 0.98	
PBT24-50 µM TMZ	6	22.182	19.214	2.968 ± 0.11	−0.584
SF8628-control	6	23.449	19.017	4.432 ± 0.23	
SF8628-50 µM TMZ	6	23.177	18.966	4.211 ± 0.21 ^c^	−0.221

*^a^ p* = 0.0022, compared with PBT24-control (*CDH1*); ^b^
*p* = 0.0022, compared with PBT24-50 µM TMZ (*CDH1*); ^c^
*p* = 0.0022, compared with PBT24-50 µM TMZ (*CDH2*).

## Data Availability

The data presented in this study are available on request from the corresponding author.

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
