# Peer review of "The Different Temozolomide Effects on Tumorigenesis Mechanisms of Pediatric Glioblastoma PBT24 and SF8628 Cell Tumor in CAM Model and on Cells In Vitro"

_ijms, 2022, doi:10.3390/ijms23042001_

Round 1
Reviewer 1 Report
The paper by Damanskienė and coworkers highlighted the differences in TMZ treatment by using two different glioma cell lines and CAM model suggesting tha TMZ sensitivity is necessary before treatment. The paper is interesting; however, some point should be improved and/or clarified.
-Authors declared in material and methods section that the 6 groups are PBT24-control (n = 13), PBT24-50 μM TMZ (n = 13), PBT24-100 μM TMZ (n = 10), SF8628-control (n = 13), SF8628-50 μM TMZ (n = 14), SF8628-100 μM TMZ (n = 13). However, then they did not shown results using the same number of samples. Authors should clarify.
-In addition to PCNA, it would be interesting to evaluate an apoptotic marker.
-Given that treatment was performed by using 50 and 100 mM of TMZ, why Slc12a2, Slc12a5, CDH1 and CDH2 expression was only analyzed in 50 mM of TMZ treatment?
-In figure 3 data are shown without standard deviation. Figure should be revised.
-Real time PCR results should be shown in a clearer way. Authors should display the data as folds respect to the control. Thus, changes induced by the TMZ treatment should be shown as fold respect to control chosen as calibrator (=1).
-Authors evaluated the expression of CDH1. Given that controversial data are present about its levels of expression in gliomas and its involvement in glioma progression and invasion (doi.org/10.1371/journal.pone.0013665; doi.org/10.3390/biomedicines910132), authors should discuss this point in the Discussion section where detailed information about the results should be removed.
-In the abstract section authors declared that the study supports the suggestion that individual sensitivity to TMZ should be assessed before the beginning of the treatment. Authors should improve the Discussion by summarizing at the end what are the differences between the two pediatric glioma lines that are associated with dissimilar TMZ response.
Author Response
-Authors declared in material and methods section that the 6 groups are PBT24-control (n = 13), PBT24-50 μM TMZ (n = 13), PBT24-100 μM TMZ (n = 10), SF8628-control (n = 13), SF8628-50 μM TMZ (n = 14), SF8628-100 μM TMZ (n = 13). However, then they did not shown results using the same number of samples. Authors should clarify.
Answer
The group samples size you refer to in the section "4.3. The PBT24 and SF8628 tumor study groups" are the same to those listed in Table 1. Larger sample sizes are needed to track tumor invasion frequency. The samples in the immunohistochemistry groups reported in section 4.3 are consistent with the samples reported in Table 2. The immunohistochemistry sample number was lower than those of the groups investigating tumor invasion. A smaller sample size was sufficient to assess immunohistochemistry data. Another reason for the difference in samples for immunohistochemical markers, for example, between PCNA and EZH2, is that more sections are needed to detect the presence of invasion in some cases. The tumors are small, so we have limited ourselves to a smaller sample size of immunohistochemistry groups, which ensures the reliability of the data and allows us to perform the other immunohistochemistry studies foreseen in the project (we have also investigated more cancer markers which data have not shown in the paper).
For clarity, we have added the sample (n) to Tables 3 and 4 for the gene expression studies.
-In addition to PCNA, it would be interesting to evaluate an apoptotic marker.
Answer
We have added to the text of the manuscript Discussion accordingly, and we think this adequately meets the requirement of the comment.
-Given that treatment was performed by using 50 and 100 mM of TMZ, why Slc12a2, Slc12a5, CDH1 and CDH2 expression was only analyzed in 50 mM of TMZ treatment?
Answer
In choosing the 50 uM dose to treat cells for gene expression studies, we were guided by the concentration of TMZ produced in the plasma of glioblastoma patients. The literature states: "The plasma concentration is known to follow a dose-dependent pattern as the administration of a dose of between 150-200 mg/m2/day resulted in peak concentrations of 3 to 15 μg/ml (15-77 μM).”
Thank you very much for your comment. We have updated the Methods section (4.9) to cite the source [64].
-In figure 3 data are shown without standard deviation. Figure should be revised.
Answer
Figure 3 shows the percentage of the tumor invasion, so there is no standard deviation. We believe that Figure 3 is relevant.
-Real time PCR results should be shown in a clearer way. Authors should display the data as folds respect to the control. Thus, changes induced by the TMZ treatment should be shown as fold respect to control chosen as calibrator (=1).
Answer
Thank you for your comment. We have fulfilled the requirement by replacing Figure 8.
-Authors evaluated the expression of CDH1. Given that controversial data are present about its levels of expression in gliomas and its involvement in glioma progression and invasion (doi.org/10.1371/journal.pone.0013665; doi.org/10.3390/biomedicines910132), authors should discuss this point in the Discussion section where detailed information about the results should be removed.
Answer
Thank you very much for your critical comment. We have appropriately revised the Discussion text in line with the note. We were unable to open the link doi.org/10.3390/biomedicines910132. However, we understood the comment and hoped we corrected the text appropriately.
-In the abstract section authors declared that the study supports the suggestion that individual sensitivity to TMZ should be assessed before the beginning of the treatment. Authors should improve the Discussion by summarizing at the end what are the differences between the two pediatric glioma lines that are associated with dissimilar TMZ response.
Answer
The corrections to the comment are made.
Note: Conclusions are adjusted
We thank the reviewer for his valuable comments, which have helped to improve the manuscript.

Reviewer 2 Report
The authors demonstrated the temozolomide effects on tumorigenesis mechanisms of pediatric glioblastoma PBT24 and SF8628 cell tumors using a CAM model. A CAM model is very unique and useful to investigate the effects of anticancer medication. However, the following issues should be clarified to improve this manuscript and to accept this article:
- The authors had better mention the benefits of this CAM model compared to 3D culture systems and the mouse models using immunodeficient mice.
- Why did the authors focus on EZH2 and cadherins in GBM cells? Moreover, what is the relationship between these molecules and ion transporters (NKCC and KCC)? The authors had better explain them and add some experiments if necessary.
Author Response
Response to Reviewer 2 Comments
The authors demonstrated the temozolomide effects on tumorigenesis mechanisms of pediatric glioblastoma PBT24 and SF8628 cell tumors using a CAM model. A CAM model is very unique and useful to investigate the effects of anticancer medication. However, the following issues should be clarified to improve this manuscript and to accept this article:
The authors had better mention the benefits of this CAM model compared to 3D culture systems and the mouse models using immunodeficient mice.
Answer
We have made the appropriate corrections to the Introduction text. We hope that we have fulfilled the requirement of the comment.
Why did the authors focus on EZH2 and cadherins in GBM cells? Moreover, what is the relationship between these molecules and ion transporters (NKCC and KCC)? The authors had better explain them and add some experiments if necessary.
Answer
Thank you for your comment. We are investigating the effect of other drugs on EZH2 in adult glioblastoma in relation to gender, so it was important for us to assess the impact of TMZ so that we can then perform comparative studies. We agree that the significance of the cadherins studied is debatable, but there are only a few studies of them in phGBM in literature. Therefore, we believe that it is appropriate to perform these studies. According to the comment, we have made the proper changes to the text in the link on cadherins, and we believe these changes have improved the manuscript.
NKCC1 and KCC2 are separate co-transporters that are very important for intracellular chloride regulation, and their evaluation requires parallel studies of both carriers. The assays performed allow a sufficient assessment of the differences between the cells tested and received data are important to evaluating the possible ineffectiveness of TMZ. We believe that the data presented are adequate for the manuscript.
We thank the reviewer for his valuable comments, which have helped to improve the manuscript.

Reviewer 3 Report
Dear Authors,
Your manuscript on the different temozolomide effect on tumorogenesis mechanisms of pediatric glioblastoma PBT24 and SF8628 cell tumor in CAM model and on cells in vitro is an interesting approach for treatment analysis. Reliable models in glioma research are urgently needed and therefore I consider your manuscript a valuable contribution to this field. Nevertheless, I do have some remarks that should be considered to improve the manuscript.
- Most of the literature that you cite deals with adult glioblastoma. In this group the heterogeneity is very high, but the genetically diversity in pediatric tumors is even higher. Both groups share only a very limited overlap. Therefore the pediatric tumors should not be compared to adult counterparts. I would recommend to concentrate on literature on pediatric tumors describing the distinct tumor type you use in your experiments.
- Your introduction is very comprehensive but to my impression it would benefit from a clearer focus / storyline. To me the connection of the different findings you want to analyze in your experimental design is not put into the focus of the introduction. In addition I would recommend to write some of the statements less strong. Especially in line 86-89 you refer to two publications and your text sounds to me that this is well known scientific fact. To me this a bit overinterpreted. You can either cite more articles, a review of this topic or relativize your statement.
- The data you present on your CAM model would benefit from a better visualization. CAM thickness for example would definitely benefit from showing it in a box plot, as the spread is quite huge. The infiltration percentage needs some more explanation and the figure 3 should be shown with error bars.
- Please explain why you did stop using 100 µM TMZ in all expression experiments. Based on the previous data there you should detect the even more pronounced effects.
Minor remarks
- Figure 4 seems to have different font sizes.
- In the text and table 3 the writing style of the genes implements that these genes are not human. Human gene names should be written in capital letters and in italics. The writing currently implements the use of mouse genes.
- In line 322 you mention the ratio of SLCs, which only plays a minor part in the corresponding text. I would recommend to put this into a figure, if it is important for the discussion.
Best regards
Author Response
Response to Reviewer 3 Comments
Your manuscript on the different temozolomide effect on tumorogenesis mechanisms of pediatric glioblastoma PBT24 and SF8628 cell tumor in CAM model and on cells in vitro is an interesting approach for treatment analysis. Reliable models in glioma research are urgently needed and therefore I consider your manuscript a valuable contribution to this field. Nevertheless, I do have some remarks that should be considered to improve the manuscript.
Most of the literature that you cite deals with adult glioblastoma. In this group the heterogeneity is very high, but the genetically diversity in pediatric tumors is even higher. Both groups share only a very limited overlap. Therefore the pediatric tumors should not be compared to adult counterparts. I would recommend to concentrate on literature on pediatric tumors describing the distinct tumor type you use in your experiments.
Answer
We agree with the observation that pediatric and adult glioblastomas cannot be directly compared. We have avoided making such a comparison in this manuscript. We have added to the introduction a reference to the fact that the treatment of children cannot be directly applied to the treatment of adults. As there are minimal studies in the literature on the cell line tumors that we have studied, we have inevitably had to rely on some data from adult studies. We hope that the additions we have made are more focused on the specifics of the treatment and have, therefore, at least partially fulfilled the requirement of the comment.
Your introduction is very comprehensive but to my impression it would benefit from a clearer focus / storyline. To me the connection of the different findings you want to analyze in your experimental design is not put into the focus of the introduction. In addition I would recommend to write some of the statements less strong. Especially in line 86-89 you refer to two publications and your text sounds to me that this is well known scientific fact. To me this a bit overinterpreted. You can either cite more articles, a review of this topic or relativize your statement.
Answer
In light of the comment, we have added text to the article's storyline to address the specificities of pediatric glioblastoma treatment. We have devoted more space to the importance of the chloride cotransporter assays in this study and the need for a parallel study of NKCC1 and KCC2. We believe that the data warrant a more extensive further studies.
Thank you for your comment on the overinterpretation of the CAM method. We hope that the corrections made to the text have solved this problem.
The data you present on your CAM model would benefit from a better visualization. CAM thickness for example would definitely benefit from showing it in a box plot, as the spread is quite huge. The infiltration percentage needs some more explanation and the figure 3 should be shown with error bars.
TMZ inhibited the thickening of the CAM membrane of PBT24 tumor in PBT24 tumor (treated with 100 uM) only. There are numerous figures in the manuscript, so we think it is not necessary to put an additional figure because of the repetition of data presentation.
Regarding Figure 3, the figure shows the percentage of tumor invasion, so there is no standard deviation. We believe that Figure 3 is relevant.
Please explain why you did stop using 100 µM TMZ in all expression experiments. Based on the previous data there you should detect the even more pronounced effects.
Answer
In choosing the 50 μM dose to treat cells for gene expression studies, we were guided by the concentration of TMZ produced in the plasma of glioblastoma patients. The literature states: "The plasma concentration is known to follow a dose-dependent pattern as the administration of a dose of between 150-200 mg/m2/day resulted in peak concentrations of 3 to 15 μg/ml (15-77 μM).
Thank you very much for your comment. We have updated the Methods section (4.9) to cite the source [64].
Minor remarks
Figure 4 seems to have different font sizes.
Answer
Thanks for noticing. We have corrected the discrepancy.
In the text and table 3 the writing style of the genes implements that these genes are not human. Human gene names should be written in capital letters and in italics. The writing currently implements the use of mouse genes.
Answer
Thank you very much for your note. We have corrected this error.
In line 322 you mention the ratio of SLCs, which only plays a minor part in the corresponding text. I would recommend to put this into a figure, if it is important for the discussion.
Answer
Thank you very much for your comment and recommendation. We have added Figure 8 to the results section.
Note: Conclusions are adjusted.
The English of the text has been checked and corrected.
We thank the reviewer for his valuable comments, which have helped to improve the manuscript.

Round 2
Reviewer 1 Report
The manuscript has been improved. However, the statistical analysis of real time PCR data should be perfected in figure 6 table 3 and figure 8 table 4 because standard deviation is missing. Authors show the mean of PCR results without standard deviation.
Moreover it is not clear what the authors added in the text about the question on apoptotic marker.
Author Response
(x) English language and style are fine/minor spell check required
Answer
Requirement fulfilled, text revised and corrected.
Comments and Suggestions for Authors
The manuscript has been improved. However, the statistical analysis of real time PCR data should be perfected in figure 6 table 3 and figure 8 table 4 because standard deviation is missing. Authors show the mean of PCR results without standard deviation.
Answer
Thank you for your repeated comment. When we made the corrections the first time, we did not fully understood the requirement. We have now entered the PCR data in Tables 3 and 4 in Figures 6 and 8 with the mean with errors. We have enlarged the graphs slightly and made the font size the same. Also, in the descriptions of the graphs, the median with min and max values was incorrectly stated in the primary variable. The same was true for Graph 7.
We are very grateful to Reviewer for the repeated comment, which allowed us to avoid errors.
Moreover it is not clear what the authors added in the text about the question on apoptotic marker.
-In addition to PCNA, it would be interesting to evaluate an apoptotic marker.
Answer
We apologise for not providing point-by-point corrections to the addition of references after the first revision.
References No. 7, 13, 29, 30, 31, 43, 65 have been added since the original version, taking into account the comments of the reviewers.
Additional sources added related to the PCNA are No. 13 (Introduction) and No. 13 and No. 43 (Discussion).
For cadherins, additional references No. 29, 30 and 31 (Introduction and Discussion) have been added.
Reviewer 2 Report
This manuscript is improved and is acceptable now. Congratulations!
Author Response
Comments and Suggestions for Authors
This manuscript is improved and is acceptable now. Congratulations!
Answer
Thanks again for your comments, which have helped to improve the manuscript.
Reviewer 3 Report
Dear Authors,
Thank you for addressing all comments adequately. I would only like to point out that in figure 4, 6 and 8 I do see a blue line in the middle of the figure. I suppose it is there due to editing and will disappear in the final version, but recommend to check this point in the proof.
Best regards
Author Response
(x) English language and style are fine/minor spell check required
Answer
Requirement fulfilled, text revised and corrected.
Comments and Suggestions for Authors
Thank you for addressing all comments adequately. I would only like to point out that in figure 4, 6 and 8 I do see a blue line in the middle of the figure. I suppose it is there due to editing and will disappear in the final version, but recommend to check this point in the proof.
Answer
In the clean version, figures 4, 6, 8 are transparent.
Thanks again for your comments, which have helped to improve the manuscript.